# Time-Dependent Proteomic Signatures Associated with Embryogenic Callus Induction in *Carica papaya* L.

**DOI:** 10.3390/plants12223891

**Published:** 2023-11-18

**Authors:** Lucas Rodrigues Xavier, Caio Cezar Guedes Corrêa, Roberta Pena da Paschoa, Karina da Silva Vieira, Daniel Dastan Rezabala Pacheco, Lucas do Espirito Santo Gomes, Bárbara Cardoso Duncan, Laís dos Santos da Conceição, Vitor Batista Pinto, Claudete Santa-Catarina, Vanildo Silveira

**Affiliations:** 1Laboratório de Biotecnologia, Centro de Biociências e Biotecnologia (CBB), Universidade Estadual do Norte Fluminense Darcy Ribeiro (UENF), Campos dos Goytacazes 28013-602, RJ, Brazil; rxlucas@outlook.com (L.R.X.); caiocagronomo@gmail.com (C.C.G.C.); robertappaschoa@pq.uenf.br (R.P.d.P.); vieirak298@gmail.com (K.d.S.V.); danielpdastan@gmail.com (D.D.R.P.); lucas.esgomes19@gmail.com (L.d.E.S.G.); 20211200075@pq.uenf.br (B.C.D.); lais_santosc@hotmail.com (L.d.S.d.C.); 2Unidade de Biologia Integrativa, Setor de Genômica e Proteômica, Universidade Estadual do Norte Fluminense Darcy Ribeiro (UENF), Campos dos Goytacazes 28013-602, RJ, Brazil; 3Laboratório de Biologia Celular e Tecidual, Centro de Biociências e Biotecnologia (CBB), Universidade Estadual do Norte Fluminense Darcy Ribeiro (UENF), Campos dos Goytacazes 28013-602, RJ, Brazil; vitorbp@uenf.br (V.B.P.); claudete@uenf.br (C.S.-C.)

**Keywords:** micropropagation, bottom-up proteomics, *Carica papaya*, embryogenic callus, somatic embryos, time-series analysis

## Abstract

Sex segregation increases the cost of *Carica papaya* production through seed-based propagation. Therefore, in vitro techniques are an attractive option for clonal propagation, especially of hermaphroditic plants. Here, we performed a temporal analysis of the proteome of *C. papaya* calli aiming to identify the key players involved in embryogenic callus formation. Mature zygotic embryos used as explants were treated with 20 μM 2,4-dichlorophenoxyacetic acid to induce embryogenic callus. Total proteins were extracted from explants at 0 (zygotic embryo) and after 7, 14, and 21 days of induction. A total of 1407 proteins were identified using a bottom-up proteomic approach. The clustering analysis revealed four distinct patterns of protein accumulation throughout callus induction. Proteins related to seed maturation and storage are abundant in the explant before induction, decreasing as callus formation progresses. Carbohydrate and amino acid metabolisms, aerobic respiration, and protein catabolic processes were enriched throughout days of callus induction. Protein kinases associated with auxin responses, such as SKP1-like proteins 1B, accumulated in response to callus induction. Additionally, regulatory proteins, including histone deacetylase (HD2C) and argonaute 1 (AGO1), were more abundant at 7 days, suggesting their role in the acquisition of embryogenic competence. Predicted protein–protein networks revealed the regulatory role of proteins 14-3-3 accumulated during callus induction and the association of proteins involved in oxidative phosphorylation and hormone response. Our findings emphasize the modulation of the proteome during embryogenic callus initiation and identify regulatory proteins that might be involved in the activation of this process.

## 1. Introduction

Understanding the molecular processes that control the embryogenic switch in plant somatic cells is of interest to current plant biotechnology. In this context, the costs of *Carica papaya* production through seed-based propagation are increased by sex segregation, and in vitro tissue culture are emerging as an attractive option for clonal propagation of hermaphroditic papaya plants. In this context, it is already known that using molecular markers to select sex-determined papaya plants can increase orchard yields by up to 49% when compared to the conventional method of sex determination by flowering [1]. Thus, somatic embryogenesis (SE) in papaya can be applied to produce clones on a large-scale, reducing production costs due to the sexual segregation presented by seedlings [2]. The SE in *C. papaya* already has a well-established and characterized protocol able to generate globular, heart, torpedo, and cotyledonary embryo stages, similar to those of zygotic embryogenesis [3]. Moreover, SE in papaya also serves as an efficient experimental model for genetic, physiological, and morphological studies of the formation and development of somatic embryos in plants [4,5].

SE is a morphogenetic process in which plant somatic cells undergo transcriptomic reprogramming in response to an induction signal, primarily following auxin or stress treatments, and then start to form embryos asexually. This reprogramming prompts the cells to enter the developmental embryogenic pathway, leading to the formation of structures resembling embryos, known as somatic embryos [6]. The formation of somatic embryos can occur directly in the explant or indirectly with an intermediary callus phase [7]. Only a few of the molecular events that govern embryogenic callus induction in somatic plant cells have begun to be decoded, and with the use of global transcriptome and proteomic analysis, many genes and proteins have been shown to be involved in the SE that is induced in different plant species [8,9]. On the other hand, studies on indirect SE usually use already differentiated callus, leaving gaps in the regulatory process that directs the initiation of embryogenic callus formation.

The molecular pathways involved in the embryogenic response during in vitro morphogenesis are of particular interest in the field of plant development biology. In this way, SE studies contribute to the understanding of the regulatory mechanisms and control of totipotency and pluripotency in plant somatic cells [6]. Comparative proteomics from embryogenic calli is an interesting approach allowing the identification of proteins with relevant roles during the SE process in papaya. In this context, a proteomics study using LED lamps with different wavelengths of light during the maturation of papaya somatic embryos revealed 522 proteins, where the optimization of embryo development was associated with the accumulation of energy metabolism proteins [4]. Another study identified 801 proteins at different stages of somatic embryo development, showing specific proteomic profiles for each stage [10]. Furthermore, the study of the mitochondrial proteome of callus with contrasting embryogenic competence led to the identification of 237 proteins and contributed to the discovery of proteins important for energy production and auxin homeostasis [11]. On the other hand, these studies used already differentiated callus for proteomic analyses, where the process of induction and formation of embryogenic callus is still poorly understood. In this sense, proteomics is considered a promising technique to improve our knowledge of the physiological and molecular factors that affect the plant SE process [9].

Our hypothesis is that specific protein sets function at different stages of embryogenic callus induction in order to coordinate this process. To address this question, we performed a time-course analysis of the proteomic responses during the onset of embryogenic callus formation in papaya. Our findings highlighted that induction treatment triggers hormonal responses and epigenetic modulation, leading to the accumulation of regulatory proteins for the initiation of embryogenic callus. These data can provide insights into alternatives to SE induction in papaya.

## 2. Results

### 2.1. Embryogenic Callus Induction

Upon removal of the seed tegument, the cotyledonary zygotic embryo used as explant exhibited a white color (T0; Figure 1A). The cotyledons bent after 7 days of inoculation (T7) in the induction treatment, while callus formation was observed 14 days after inoculation (T14; Figure 1B,C). After 21 days (T21), the embryogenic callus had already formed up to the entire explant (Figure 1D). During callus induction, the formation of a mass of yellowish and brownish cells was observed. In addition, callus formation is also characterized by an increase in cell mass and growth of the explant in the induction culture medium.

### 2.2. Proteomics Analysis during Embryogenic Callus Induction

To understand the molecular signature of embryogenic callus formation, a time-series bottom-up proteomic approach was performed. A total of 1407 proteins were identified in bottom-up proteomic analysis of explant samples before and at 7, 14, and 21 days after induction of embryogenic callus (Appendix A). After the comparative proteomic analysis, 957 proteins presented differential accumulation in at least one comparison between the different induction times (Appendix A). The principal component analysis (PCA) showed a separation between explants before (zygotic embryo) the other sampling times in the first component (Appendix A). In addition, callus induction at 7 days was equidistant between explants before incubation and callus after 14 and 21 days of incubation. However, explants after 7 days of induction showed a difference from the other treatments observed in the second component, probably due to a group of proteins responsible for initiating the process of dedifferentiation. The results show that callus cells at 14 and 21 days were closely grouped, possibly indicating that after 14 days of incubation, the effects of the induction treatment are smaller.

The clustering analysis revealed four clusters with distinct patterns of protein accumulation throughout embryogenic callus induction (Figure 2). Cluster 1 contained 386 proteins that increased accumulation at all analyzed times during induction. In contrast, cluster 2 contained 165 proteins showing decreased abundance during callus induction. Cluster 3 contained 251 proteins that were most abundant at the start of callus induction at 7 days of incubation and cluster 4 contained 155 proteins that accumulated after callus formation at 21 days of incubation (Figure 2; Appendix A). By differential accumulation analysis, it was also possible to identify proteins that have been associated with SE, and may be involved in embryogenic competence acquisition in the 2,4-D-mediated induction of *C. papaya* embryogenic callus (Table 1).

### 2.3. Functional analysis of Proteomic Data

The functional analysis revealed that proteins related to seed maturation and storage reserves were more abundant in the explants before incubation, and their accumulation progressively decreased during induction treatment (Figure 3A). On the other hand, biological processes involved in carbohydrate and amino acid metabolism, aerobic respiration and protein catabolic processes were enriched at seven days of incubation (Figure 3A). Furthermore, the cellular components of secretory vesicles and lipid droplets were enriched among the most accumulated proteins in explants before incubation (0 day), while components such as the proteasome, mitochondria, nucleolus, and cytosol were enriched among the proteins accumulated at 7, 14, and 21 days of incubation (Figure 3B).

The evaluation of the molecular functions of the differentially accumulated proteins showed enrichment of nutrient reservoirs in the explants before incubation (0 day), while functions involved in energy metabolism and stress response were enriched after the onset of induction (Figure 3C). In addition, enrichment analysis showed pathways associated with amino acid metabolism, reserve consumption, and ATP production during callus induction, indicating a metabolic shift associated with the process of embryogenic competence acquisition (Figure 3D).

### 2.4. Identification of Regulatory Proteins Differentially Accumulated during Induction

Due to the important regulatory role of protein kinases and transcription factors, a classification of the identified proteins was performed based on already known classes/types (Appendix A). As a result, all differentially accumulated regulatory proteins were more abundant after the start of induction treatment, and no protein kinase or transcriptional regulator was identified in the explant before incubation, at 0 day. The identified regulatory proteins belong to seven different classes: CAMK_CDPK, MADS-MIKC, C2H2, CMGC_CDK-CRK7-CDK9, Whirly, CSD, and MBF1.

### 2.5. Prediction of Protein-Protein Interaction Networks

We used the data from functional analysis and differential accumulation to determine a set of proteins that may play a role in embryogenic callus formation (Table 1). The FASTA sequences of *C. papaya* proteins were used in STRING to predict protein–protein interactions. The search resulted in the identification of networks associated with oxidative phosphorylation, hormone response, SAM metabolism, and regulatory proteins 14-3-3 (Figure 4). In this network, the 14-3-3 proteins play an important role by connecting interactions between phosphatases, proteins associated with hormonal responses, and proteins involved in metabolic activity.

## 3. Discussion

### 3.1. The Explant Contains Storage Reserves That Are Consumed during the Induction of Embryogenic Callus through Changes in Energy Metabolism

During zygotic embryogenesis, proteins and lipids accumulate throughout development [12]. Here, we identified proteins that accumulated in the explants, i.e., zygotic embryos before incubation (T0 days), that were enriched in the biological process of seed maturation (Figure 3A). Many of them have a nutrient reserve role, and their accumulation decreased over the course of embryogenic callus induction (Figure 1C). Our data reveal proteins involved in nutrient reservoir activity, such as 12S seed storage protein (evm.model.supercontig_165.8) and legumin B-like (evm.model.supercontig_165.7), which are transcriptionally active in embryogenic cultures and modulated in response to ABA [13,14]; oleosin 5-like (evm.model.supercontig_110.15) and oleosin 1 (evm.model.supercontig_13.63); and late embryogenesis abundant (LEA) proteins, such as 18 kDa seed maturation protein (evm.model.supercontig_85.72) and LEA protein 31 (evm.model.supercontig_58.99). An enhanced accumulation of storage reserves in embryogenic callus seems to be a precursor to the successful differentiation of somatic embryos [10]. Cells utilize stored carbohydrates, proteins, and other nutrients as a source of energy and building blocks for cell division and growth. During embryogenic callus induction, ABA initiates reprogramming of somatic cells, while LEA proteins protect cells from stress [15,16].

Modulation of embryogenic competence in *C. papaya* callus involves the accumulation of energy metabolism-related proteins [11]. In this regard, proteins with a role in electrochemical gradient generation and proton transport were identified, as well as pyrophosphatases regulated throughout callus induction (Table 1). Additionally, these proteins have predicted interactions that may be related to energy generation (Figure 3). This result is in agreement with the observation that biological processes such as protein catabolism and carbohydrate and amino acid metabolisms are enriched among DAPs after the start of embryogenic callus formation (Figure 3A). Furthermore, predicted interactions between oxidative phosphorylation proteins and 14-3-3 proteins were also observed, suggesting that they play an important regulatory role in integrating hormone responses with energy metabolism during embryogenic callus formation (Figure 4). These changes in energy metabolism could be related to the effects of 2,4-D and sucrose present in the induction culture medium, in addition to the catabolism of storage molecules present in the explant, such as globulins, lipids, and carbohydrates.

### 3.2. Acquisition of Embryogenic Competence in Somatic Cells Involves Modulation of Regulatory Proteins and Epigenetic Mechanisms

Auxin-induced SE involves the modulation of epigenetic mechanisms such as DNA methylation and miRNA-directed silencing, where histone deacetylases (HDACs) and argonaute protein 1 play important roles [6]. In this context, the phenotype of different HDAC mutants of A. thaliana was studied during direct SE, leading to the identification of HDACs that participate in embryo differentiation [17]. Additionally, several studies have demonstrated that histone acetylation/deacetylation is an important regulator of the in vitro induced embryogenic transition and is related to the differential expression of several elements of the HDAC gene families and histone acetyltransferase [18,19]. The increase in the accumulation of histone deacetylase HDT1 (evm.model.supercontig_5.85, orthologous to AtHD2C) protein was accumulated throughout the induction of the embryogenic callus (Appendix A). Studies in sugarcane have shown that histone deacetylase 1, belonging to the HDAC family, is also involved in the differentiation of somatic embryos, interacting with corepressors such as TOPLESS to modulate protein post-translational regulation during somatic embryo differentiation [20]. Thus, the temporal analysis of the data in the present study suggests that histone acetylation/methylation is regulated at early stages of callus formation and acquisition of competence in *C. papaya* embryogenic callus. On the other hand, this study presents several proteins and interactions that are not observed in models of somatic embryogenesis regulation that only take into account transcriptomic and genetic studies [6,17,21]. It is clear that elucidating the regulatory mechanism involved in somatic embryogenesis depends on studying different layers of regulation of gene expression, from the epigenetic regulation of DNA, to the maturation and transport of messenger RNA, protein translation, and post-translational modifications.

Our data show that protein argonaute 1 (evm.model.supercontig_1673.2, orthologous to AtAGO1) accumulates throughout embryogenic callus induction and has predicted interactions with two SKP1-like proteins 1B (evm.model.supercontig_26.289 and evm.model.supercontig_6.295, both orthologous to AtSKP1), which are important for SCF^TIR1/AFB^ complex assembly and auxin responses, as well as SKP1-like protein 1B (evm.model.supercontig_20.89, orthologous to AtASK2) (Table 1) (Figure 4). The participation of SKP in callus induction has been suggested before, given its role in lateral root formation [7]. In this regard, the present study provides new evidence for the participation of SKP1 and ASK2 in *C. papaya* callus formation, in addition to their possible association with AGO1-mediated epigenetic regulation. In this context, DNA methylation is also a silencing mechanism known to be involved in the regulation of genes with a role in SE [22]. In the present study, S-adenosylmethionine synthase 2 (evm.model.supercontig_67.71), adenosylhomocysteinase (evm.model.supercontig_157.46), and methionine S-methyltransferase (evm.model.supercontig_173.3) proteins accumulated during embryogenic callus initiation, where they may play a role in S-adenosylmethionine (SAM) homeostasis for regulating gene transcription (Table 1) (Figure 4). These interactions may play an important role in modulating the epigenetic mechanisms associated with the cell cycle and proliferation of callus cells in *C. papaya*. In this regard, further studies should aim to investigate how 2,4-D could be associated with the accumulation of proteins involved in epigenetic regulation during the formation of embryogenic callus.

Phosphorylation is a post-translational modification involved in fine tuning the interactions, localization, activity, and shape of proteins. Several proteins with regulatory roles are phosphorylated at specific tyrosine, threonine, and serine residues in embryogenic calli and differentiating somatic embryos [20,23]. Our data reveal the accumulation of two serine/threonine-protein phosphatase PP2A-2 catalytic subunit (evm.model.supercontig_13.7 and evm.model.supercontig_1.195, orthologous to AtPP2A-2 and AtPP2A-3, respectively) and two serine/threonine-protein phosphatase 2A 65 kDa regulatory subunit A beta isoforms (evm.model.supercontig_13.7 and evm.model.supercontig_1.195, both orthologous to AtPP2AA2) throughout embryogenic callus induction and their predicted interaction with 14-3-3 proteins (Table 1) (Figure 4). The PP2A-3 protein is known to play a role in regulating stem cell division in Arabidopsis roots [24]. In contrast, PP2AA2 regulates the phosphorylation of PIN-FORMED (PIN) proteins, which are involved in auxin flux during plant development [25]. These data provide clues to understanding how protein phosphorylation is involved in embryogenic callus formation, where PP2A and PP2AA2 phosphatases may play a role. However, the targets of these phosphatases during callus induction still need to be identified to fully understand their participation in SE. Comparative phosphoproteomics studies could help identifying phosphorylation sites and motifs involved with protein kinase and phosphatase activity associated with the induction of embryogenic callus.

### 3.3. Proteins Involved in Hormone Responses Are Induced Early after Callus Induction

The regulation of endogenous auxin levels is one of the main properties leading to the use of 2,4-D for the induction of embryogenic callus [21]. During the SE of *C. papaya*, it has already been demonstrated that the use of 2,4-D in induction is essential for the formation of embryogenic callus [4]. Our data reveal that several proteins involved in hormone regulation were differentially accumulated during *C. papaya* embryogenic callus initiation. These proteins are involved in indole-3-acetic acid (IAA) conjugation and can regulate free auxin levels, as well as proteins involved in ethylene responses. Auxin homeostasis is an important factor in the embryogenic competence of *C. papaya*, where the accumulation of GH3 proteins has already been associated with the differentiation of somatic embryos [11]. In the present study, a probable indole-3-acetic acid-amido synthetase GH3.1 (evm.model.supercontig_292.1) and two GH3.6 (evm.model.supercontig_1065.2 and evm.TU.contig_32826.1, orthologs of AtDFL1) accumulated during the induction of embryogenic callus (Table 1) (Figure 4). These proteins are involved in the conjugation of amino acids to free auxin, regulating the levels of responses of this hormone [26,27]. Thus, our data indicate that GH3 proteins are unique or accumulate during embryogenic callus formation and may participate in the regulation of free auxin homeostasis, as suggested by temporal analysis.

The participation of 14-3-3 proteins in the regulation of SE has been evidenced in different studies [9,28]. Here, several 14-3-3 proteins were identified throughout the process of embryogenic callus formation (Table 1). Furthermore, these proteins have predicted interactions and are associated with hormone responses (Figure 4). The 14-3-3 proteins have a known role in modulating ethylene levels, where RCI1A and GF14 regulate 1-aminocyclopropane-1-carboxylate accumulation through interactions with 1-amino-cyclopropane-1-carboxylate synthase (ACS), impacting responses involving ethylene [29,30]. Consistently, 1-aminocyclopropane-1-carboxylate oxidase (evm.model.supercontig_152.58, orthologous to AtEFE1) accumulated during embryogenic callus induction in the present study (Table 1). In this context, it is already known that 2,4-D regulates ethylene biosynthesis and that modulation of this hormone is important for auxin responses during SE [31]. Thus, 14-3-3 proteins may also play an important role in regulating ethylene responses during the formation of *C. papaya* embryogenic calli. Interestingly, our predicted network reveals an important role of 14-3-3 proteins interacting simultaneously with protein phosphatases in auxin responses and electrochemical gradient generation, indicating their participation in the integration of these processes. Given the relevance of using sex determined seedlings for papaya cultivation, these findings could contribute to the optimization of the cloning process through somatic embryogenesis and facilitate the production of cloned plantlets [1,2].

## 4. Materials and Methods

### 4.1. Plant Material and Callus Induction

The culture medium for callus induction was prepared using Murashige and Skoog (MS) salts and vitamins (Phytotechnology Lab, Lenexa, KS, USA) [32] supplemented with 30 g L^−1^ sucrose (Vetec, São Paulo, Brazil) and 20 μM 2,4-dichlorophenoxyacetic acid (Sigma-Aldrich, St. Louis, MO, USA), with the pH adjusted to 5.8. Then, 2 g L^−1^ Phytagel^®^ (Sigma-Aldrich) was added to the medium, followed by autoclaving at 121 °C for 15 min and polymerization in Petri dishes (90 × 15 mm) on a laminar flow [10]. The mature fruits of papaya cv ‘Golden’ were washed in running water and transferred to a laminar flow hood for immersion in 70% ethanol (Sigma-Aldrich) for 1 min and 30 min in 50% commercial bleach Qboa^®^ (Anhembi SA, Osasco, SP, Brazil) containing sodium hypochlorite from 1 to 1.25%. The fruit was washed three times with autoclaved distilled water and sliced in half with a sterile knife. The seeds were dissected with a sterile scalpel, and ten mature zygotic embryos were used as explants and inoculated in Petri dishes containing callus induction culture medium. The explants were maintained in darkness at 25 ± 1 °C for up to 21 days for callus induction. Samples of explants before incubation (time 0), and after 7, 14, and 21 days of callus induction were obtained for proteomic analysis.

### 4.2. Protein Extraction and Digestion

Three biological replicates (each biological replicate constituted by a pool of 25 explants) collected during the time of incubation were pulverized in 1.5 mL microtubes until a powder. Next, 375 μL extraction buffer consisting of 7 M urea (GE Healthcare, Freiburg, Germany), 2 M thiourea (GE Healthcare), 2% Triton X-100 (GE Healthcare), 1% dithiothreitol (DTT, GE Healthcare), and 1 mM phenylmethanesulfonyl fluoride (PMSF, Sigma-Aldrich) was added to the sample powder. The protein concentration of each biological replicate was estimated using the 2-D Quant Kit (Cytiva, Marlborough, MA, USA).

The protein samples were precipitated using a methanol/chloroform method to remove any interfering compounds from the samples according to Nanjo, et al. [33]. The samples were then resuspended in a solution consisting of 7 M urea and 2 M thiourea, after which tryptic protein digestion (1:100 enzyme:protein, V5111, Promega, Madison, WI, USA) was performed using the filter-aided sample preparation (FASP) method as described in Wiśniewski, et al. [34] with changes described in Botini, et al. [10]. The peptides were vacuum-dried and solubilized in 50 μL of a solution containing 5% (*v*/*v*) acetonitrile (Thermo Fisher Scientific, Waltham, MA, USA) and 0.1% (*v*/*v*) formic acid (Sigma-Aldrich) in mass spectrometry (MS)-grade water (Sigma-Aldrich). The peptide concentration was estimated by measuring A205 nm using a NanoDrop 2000c spectrophotometer (Thermo Fisher Scientific). The peptides were stored at −80 °C until mass spectrometry analyses.

### 4.3. Bottom-Up Proteomics Analysis

A nanoAcquity UPLC M-class (Waters, Manchester, UK) coupled to a Synapt G2-Si HDMS mass spectrometer (Waters, Manchester, UK) with an electrospray ionization (ESI)-MS/MS source was used. A total of 2 μg of peptides was loaded onto a C18 trap column (100 Å, 5 µm, 180 µm × 20 mm, 2D; Waters) at 5 μL min^−1^ for 3 min and then onto an HSS T3 analytical reversed-phase column (100 Å, 1.8 µm, 75 µm × 150 mm; Waters) at 400 nL min^−1^ heated to 45 °C. For peptide elution, a binary gradient was used, with mobile phase A consisting of water and 0.1% formic acid and mobile phase B consisting of acetonitrile and 0.1% formic acid. Gradient elution was started with 5% B, which was then increased to 40% B until 92 min and from 40 to 99% B until 96 min, after which it was maintained at 99% until 100 min, posteriorly decreased to 5% B until 102 min, and maintained at 5% B until the end of the run at 118.00 min.

The mass spectrometer was set up in positive, resolution mode (V mode), and data-independent acquisition (DIA) mode, at 35,000 FWHM, with an ion mobility (HDMSE) wave velocity program starting at 800 m s^−1^ and ending at 500 m s^−1^. The transfer collision energy increased from 25 to 55 V in high-energy mode, the cone voltage was 40 V, and the capillary voltage was 2800 V. The pressure of the nano-flow gas was 0.5 bar, the purge gas flow rate was 150 L h^−1^, and the source temperature was 100 °C. For the time-of-flight (TOF) parameters, the scan time was set to 0.5 s in continuous mode, with a mass range of 50 to 2000 Da. Human [Glu1]-fibrinopeptide B at 100 fmol μL^−1^ was used as an external calibration standard, and lock mass acquisition was performed every 30 s. Mass spectra were subsequently acquired by using MassLynx 4.1 software (Waters, Manchester, UK).

### 4.4. Proteomics Data Analysis

The proteomics MS/MS spectra processing and database search were performed using ProteinLynx Global SERVER (PLGS) software v.3.02 (Waters) against the *C. papaya* ASGPBv0.4 protein bank from Phytozome (https://phytozome.jgi.doe.gov, accessed on 1 February 2023). The raw data processing settings included a low energy threshold of 150 (counts), a high-energy threshold of 50 and an intensity threshold of 750. Additionally, the analysis was performed with the following parameters: 2 missed cleavages, at least 3 fragment ions per peptide, a minimum of 7 fragment ions per protein, a minimum of 2 peptides per protein, fixed carbamidomethyl modifications, and variable oxidation and phosphoryl modifications. The false discovery rate (FDR) for peptide and protein identification was set at a maximum of 1%, with a minimum peptide length of 6 amino acids. The label-free quantification analyses were performed using the TOP3 quantification approach, followed by the multidimensional normalization process implemented within ISOQuant software v.1.7 [35]. The protein identification criteria were defined as follows: FDR of 1%, peptide score greater than 6, minimum peptide length of 6 amino acids and at least 2 peptides per protein. All the mass spectrometry proteomics data and results tables have been deposited in the ProteomeXchange Consortium [36] via the PRIDE [37] partner repository with the dataset identifier PXD046521.

To ensure the quality of the results after data processing, only proteins present in all three biological replicates or absent (for unique proteins) were considered for differential accumulation analysis. Proteins with significant Student’s *t*-test (two-tailed; equal variance; *p*-value < 0.05) were considered up-accumulated if the Log2 fold change (FC) was greater than 0.585 and down-accumulated if the Log2 FC was less than −0.585.

Automated functional annotation analysis was performed using OmicsBox software V3.0 (https://www.biobam.com), and a manual annotation was made in UniProtKB (https://www.uniprot.org, accessed on 1 February 2023). *Arabidopsis* orthologs of differentially accumulated proteins (DAPs) were used for enrichment analysis (*p*-value < 0.05) of biological processes, cellular components, molecular functions, and KEGG (Kyoto Encyclopedia of Genes and Genomes) pathways in the Metascape tool [38]. These orthologs were also used for predicted interaction networks through a STRING v11.5 search [39] and predicted kinases, transcription factors, and transcriptional regulators using iTAK software (http://itak.feilab.net/cgi-bin/itak/index.cgi) [40]. The principal component analysis, Venn diagram, and heatmap were plotted in R language V.4.3.1 [41]. The Mfuzz package [42] was used for clustering analysis of the DAPs (*p*-value <0.05 and Log2 FC > 0.585 or <−0.585) in at least one comparison between the induction times studied.

## 5. Conclusions

In this study, regulatory proteins such as HD2C, AGO1, SKP1, and phosphatases were accumulated during the acquisition of embryogenic competence in somatic papaya cells. In this regard, the transition from vegetative to embryogenic development could be linked to the regulation of these proteins, where modulating their activity or abundance could be a way of optimizing somatic embryogenesis in papaya and other plants. In addition, it was observed that the abundance of proteins associated with seed maturation and storage reserves decreased during callus induction, while proteins involved in energy metabolism increased to support callus multiplication. This reveals the important role of selecting optimal carbon sources to improve the in vitro conditions for growing embryogenic callus. In this context, protein–protein interactions should play a key role in integrating hormonal responses (including protein kinases and phosphatases associated with auxin responses) with energy metabolism and epigenetic regulation, where 14-3-3 proteins (accumulated during callus induction) appear to play a key role. Therefore, our results confirm that specific protein groups act at different moments to induce embryogenic callus differentiation. Further studies should aim to investigate the role of 2,4-D in the regulation of accumulation and interaction between auxin response proteins during embryogenic callus induction.

## Figures and Tables

**Figure 1 plants-12-03891-f001:**
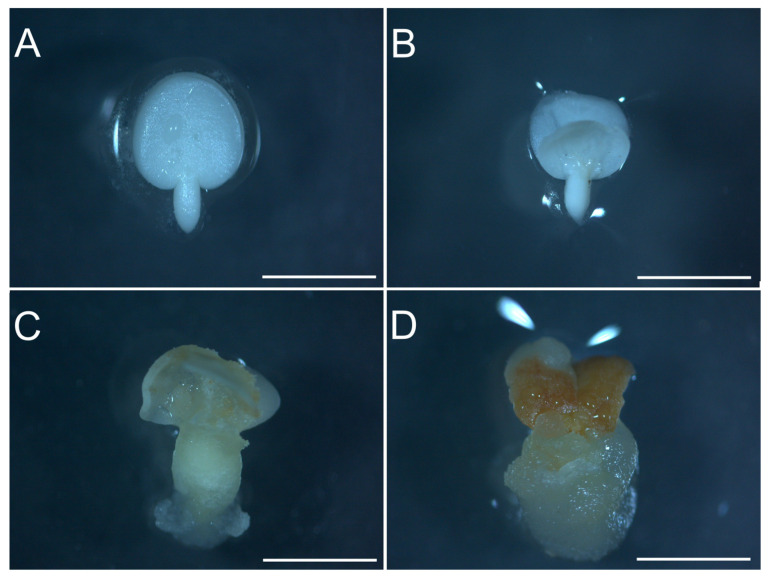
Morphological features of the explant during embryogenic callus induction using 2,4-D. Zygotic embryo used before incubation at time 0 (**A**). Initial cotlyedon folding in the induction medium after 7 days of inoculation (**B**), followed by callus formation after 14 days of inoculation (**C**), and proliferation after 21 days of inoculation (**D**). The white bar in the images represents 0.5 cm.

**Figure 2 plants-12-03891-f002:**
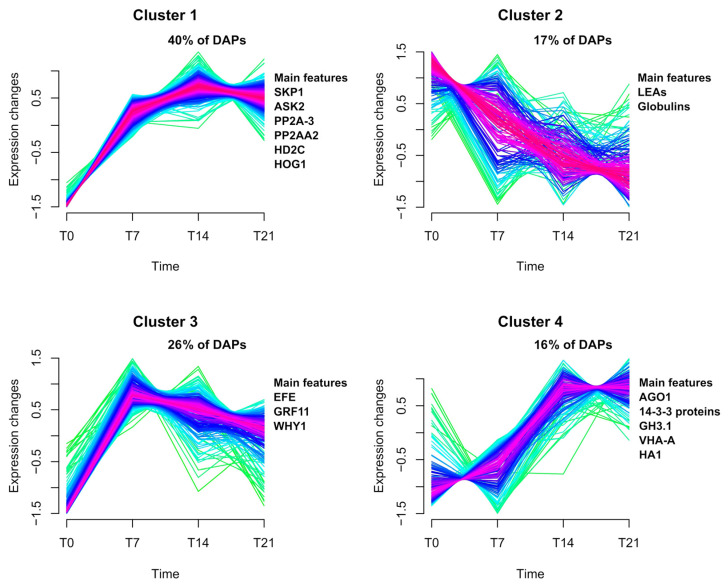
Results of the comparative analysis of proteomics data. Evaluation of the protein clusters revealed four patterns of accumulation throughout the time of callus induction. This analysis revealed proteins that increased in accumulation at all time points during callus induction (Cluster 1) and proteins that decreased their abundance throughout this process (Cluster 2). In addition, there are also groups of proteins that were abundant at the beginning of callus formation (7 days of induction) (Cluster 3), and proteins which accumulated after callus formation, at 21 days of incubation in the induction medium (Cluster 4).

**Figure 3 plants-12-03891-f003:**
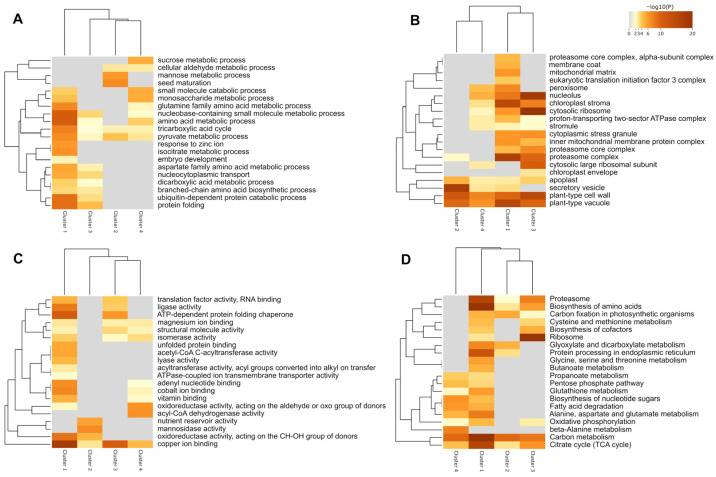
Gene ontology and KEGG pathways enrichment analysis among differentially accumulated proteins. The figure shows biological processes (**A**), cellular components (**B**), molecular function (**C**), and KEGG pathways (**D**) enriched among the different protein clusters differentially accumulated during embryogenic callus induction. The analysis shows that biological energy-generating processes are regulated during embryogenic callus induction, in addition to the accumulation of proteins with regulatory functions. In addition, several metabolic pathways are activated during embryogenic callus formation, including the metabolism of amino acids and reserve consumption.

**Figure 4 plants-12-03891-f004:**
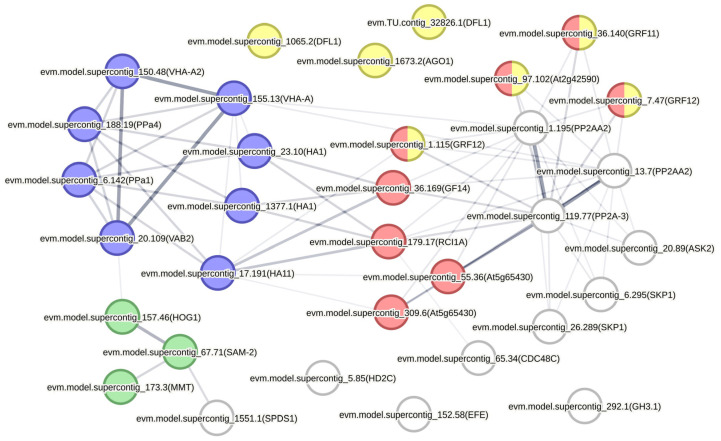
Protein–protein interaction network constructed with regulated DAPs during embryogenic callus induction. The network was constructed in STRING v.11.5 with *C. papaya* proteins. The names of the Arabidopsis orthologs are in parentheses. The color at the nodes indicates hormone response proteins (yellow), oxidative phosphorylation (blue), 14-3-3 (red), and SAM metabolism (green). The protein–protein interaction analysis highlights an important role for the 14-3-3 proteins in connecting hormonal signaling responses with energy metabolism during embryogenic callus induction. The network was constructed with medium confidence (combined score > 0.4).

**Table 1 plants-12-03891-t001:** Differentially accumulated proteins with putative association with *C. papaya* embryogenic callus induction and their respective orthologs in *A. thaliana*. After statistical and functional analysis of the comparative proteomics data, it was possible to establish a list of proteins that may play a role in the induction of embryogenic calli. The unique 7 day+ group represents proteins that were quantified at 7, 14, and/or 21 days of incubation, but were absent in all biological replicates of explants before incubation (0 day).

Protein ID	BLAST Top Hit	*Arabidopsis* Orthologs	Cluster
evm.model.supercontig_1673.2	protein argonaute 1	AGO1	4
evm.model.supercontig_20.89	SKP1-like protein 1B	ASK2	1
evm.model.supercontig_97.102	14-3-3-like protein D isoform X2	At2g42590	1
evm.model.supercontig_309.6	14-3-3-like protein B	At5g65430	4
evm.model.supercontig_55.36	14-3-3-like protein GF14 kappa isoform X2	At5g65430	4
evm.model.supercontig_65.34	cell division cycle protein 48 homolog	CDC48C	Unique_T7+
evm.model.supercontig_1065.2	indole-3-acetic acid-amido synthetase GH3.6	DFL1	Unique_T7+
evm.TU.contig_32826.1	indole-3-acetic acid-amido synthetase GH3.6	DFL1	Unique_T7+
evm.model.supercontig_152.58	1-aminocyclopropane-1-carboxylate oxidase	EFE	3
evm.model.supercontig_36.169	14-3-3-like protein	GF14	1
evm.model.supercontig_292.1	probable indole-3-acetic acid-amido synthetase GH3.1	GH3.1	4
evm.model.supercontig_36.140	14-3-3-like protein	GRF11	3
evm.model.supercontig_7.47	putative 14-3-3 protein	GRF12	4
evm.model.supercontig_1.115	14-3-3-like protein GF14 iota	GRF12	4
evm.model.supercontig_23.10	plasma membrane ATPase 4	HA1	4
evm.model.supercontig_1377.1	plasma membrane ATPase 4	HA1	Unique_T7+
evm.model.supercontig_17.191	ATPase 11, plasma membrane-type	HA11	1
evm.model.supercontig_5.85	histone deacetylase HDT1	HD2C	1
evm.model.supercontig_157.46	adenosylhomocysteinase	HOG1	1
evm.model.supercontig_173.3	methionine S-methyltransferase	MMT	Unique_T7+
evm.model.supercontig_119.77	serine/threonine-protein phosphatase PP2A-2 catalytic subunit	PP2A-3	1
evm.model.supercontig_13.7	Serine/threonine-protein phosphatase 2A 65 kDa regulatory subunit A beta isoform	PP2AA2	1
evm.model.supercontig_1.195	serine/threonine-protein phosphatase 2A 65 kDa regulatory subunit A beta isoform	PP2AA2	1
evm.model.supercontig_6.142	soluble inorganic pyrophosphatase-like	PPa1	Unique_T7+
evm.model.supercontig_188.19	soluble inorganic pyrophosphatase 4	PPa4	Unique_T7+
evm.model.supercontig_179.17	14-3-3-like protein A	RCI1A	1
evm.model.supercontig_67.71	S-adenosylmethionine synthase 2	SAM-2	2
evm.model.supercontig_26.289	SKP1-like protein 1B	SKP1	1
evm.model.supercontig_6.295	SKP1-like protein 1B	SKP1	Unique_T7+
evm.model.supercontig_1551.1	spermidine synthase-like	SPDS1	Unique_T7+
evm.model.supercontig_20.109	V-type proton ATPase subunit B 2	VAB2	1
evm.model.supercontig_155.13	V-type proton ATPase catalytic subunit A	VHA-A	4
evm.model.supercontig_150.48	V-type proton ATPase subunit a3	VHA-A2	1

## Data Availability

The data presented in this study are openly available in PRoteomics IDEntifications (PRIDE) under accession number PXD046521.

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
