# Peer review of "Time-Dependent Proteomic Signatures Associated with Embryogenic Callus Induction in *Carica papaya* L."

_plants, 2023, doi:10.3390/plants12223891_

Round 1

Reviewer 1 Report

Comments and Suggestions for Authors

In this manuscript (plants-2721832) entitled "Time-dependent proteomic signatures associated with embryogenic callus induction in Carica papaya L." submitted to Plants, Lucas Rodrigues Xavier and colleagues have performed a temporal analysis of the proteome of C. papaya calli aiming to identify the key players involved in embryogenic callus formation. This research is interesting and convincing, but minor points need to be addressed to improve the quality of this manuscript.

1. For Figure 1, at least three representative explants should be shown for morphological features of the explant during embryogenic callus induction in the revised Figure. In addition, the picture quality should be improved to show more details in the revision.

2. For Figure 2, picture of total protein separation by 1-D gel should be shown in the revised Figure

3, For Figure S2, information derived from this figure is important. Authors should consider to place this Figure into main Figures in the revised manuscript.

4, Methods for the quantitative proteomics employed in this study should be described in details in the revised section of Methods and Materials.

Author Response

Thank you very much for taking the time to review this manuscript. Please find the detailed responses attached and the corresponding revisions/corrections highlighted/in track changes in the re-submitted files

Reviewer 2 Report

Comments and Suggestions for Authors

Dear author(s),

there are some inspiring insights thorough the manuscript and I tend to agree on its publication. However, there are few points that can be quickly addressed to improve its overall communication:

Abstract:

1/ you are pointing out in the very first sentence that it is "the cost of Carica papaya production" that motivates your research hypothesis, therefore it is appropriate to mention whether you have achieved any cost reduction (quantify, or at least indicate % change)

Introduction:

2/ make sure that this chapter fully introduces any reader into to the topic, explain all the terms, jargon, abbreviations, Latin and Greek letters, and the whole context that is necessary for anyone (including experts from other disciplines) to understand the following chapters

3/ the need to reduce the production cost needs to be strengthened, provide deeper insights to investors' perspectives and refer to papers "The analysis of investment into industries based on portfolio managers" and "The Dynamic Effect of Micro-Structural Shocks on Private Investment Behavior"

4/ go straight to the point and more in depth, write more technically (always provide corresponding numbers), significantly condensate all the text by reducing ballast phrases and cliché

5/ better justify the urgency to confirm or deny the research hypothesis, the research hypothesis could be stated more clearly, condensate the research hypothesis into 1 short statement (or question) that will be subsequently confirmed or refuted, make sure the urgency and significance of the research hypothesis was justified in its environmental - economic nexus

Results:

6/ avoid data overkill, present only the most most industrially important results with a preference for those that are easier to interpret economically (in terms of cost breakdown)

7/ each Tab. and Fig. should be provided with caption that describes A/ what can be seen and B/ how is this relevant to the research hypothesis

8/ kindly note that % were designed to avoid the use of decimal symbol, "40.33%" = 40 % (likewise thorough the entire manuscript)

Discussion:

9/ once your main argument is based on cost reduction you should not ignore economic reality in the rest of the manuscript, provide at leas some simplified economic considerations

10/ propose some improvements and direction for future research, refer to paper "Silica nanoparticles from coir pith synthesized by acidic sol-gel method improve germination economics"

11/ compare your results in more depth with the existing literature, identify the main deviations and try to explain the mechanisms by which they may have been caused

Conclusions:

12/ do not repeat what was investigated, do not repeat your methods and results again and again, please understand that the Conclusion chapter is not a summary of your work (such as Abstract), present only original, generalized and industrially significant revelations that have the potential to expand the horizon of human knowledge (higher level of generalization is mandatory)

13/ clearly indicate whether the research hypotheses tends to be confirmed or not and whether the concept seems to be less costly

14/ reveal the main driving mechanisms of your results, provide deeper synthesis and reveal some more original/significant findings

Author Response

(The authors gave the same response as above.)
